# Compromised NHE8 Expression Is Responsible for Vitamin D-Deficiency Induced Intestinal Barrier Dysfunction

**DOI:** 10.3390/nu15224834

**Published:** 2023-11-19

**Authors:** Yaoyu Guo, Yanni Li, Zeya Tang, Chong Geng, Xiaoxi Xie, Shuailing Song, Chunhui Wang, Xiao Li

**Affiliations:** 1Department of Gastroenterology and Hepatology, West China Hospital, Sichuan University, Chengdu 610041, China; 13919103006@163.com (Y.G.); 17338632315@163.com (Y.L.); 15340574085@163.com (X.X.); ssljjy123@163.com (S.S.); 2Department of Outpatient, West China Hospital, Sichuan University, Chengdu 610041, China; 13882086016@163.com; 3Laboratory of Gastroenterology and Hepatology, West China Hospital, Sichuan University, Chengdu 610041, China; gengchong15scu@163.com

**Keywords:** vitamin D, vitamin D receptor, Na^+^/H^+^ exchanger isoform 8, ulcerative colitis, intestinal barrier function

## Abstract

**Objectives:** Vitamin D (VitD) and Vitamin D receptor (*VDR*) are suggested to play protective roles in the intestinal barrier in ulcerative colitis (UC). However, the underlying mechanisms remain elusive. Evidence demonstrates that Na^+^/H^+^ exchanger isoform 8 (NHE8, SLC9A8) is essential in maintaining intestinal homeostasis, regarded as a promising target for UC therapy. Thus, this study aims to investigate the effects of VitD/VDR on NHE8 in intestinal protection. **Methods:** VitD-deficient mice, *VDR*^−/−^ mice and *NHE8*^−/−^ mice were employed in this study. Colitis mice were established by supplementing DSS-containing water. Caco-2 cells and 3D-enteroids were used for in vitro studies. VDR siRNA (siVDR), VDR over-expression plasmid (pVDR), TNF-α and NF-κb p65 inhibitor QNZ were used for mechanical studies. The expression of interested proteins was detected by multiple techniques. **Results:** In colitis mice, paricalcitol upregulated NHE8 expression was accompanied by restoring colonic mucosal injury. In VitD-deficient and *VDR*^−/−^ colitis mice, NHE8 expression was compromised with more serious mucosal damage. Noteworthily, paricalcitol could not prevent intestinal barrier dysfunction and histological destruction in *NHE8*^−/−^ mice. In Caco-2 cells and enteroids, siVDR downregulated NHE8 expression, further promoted TNF-α-induced NHE8 downregulation and stimulated TNF-α-induced NF-κb p65 phosphorylation. Conversely, QNZ blocked TNF-α-induced NHE8 downregulation in the absence or presence of siVDR. **Conclusions:** Our study indicates depressed NHE8 expression is responsible for VitD-deficient-induced colitis aggravation. These findings provide novel insights into the molecular mechanisms of VitD/VDR in intestine protection in UC.

## 1. Introduction

Ulcerative colitis (UC) is a chronic and intractable inflammatory bowel disease (IBD) which has influenced millions of individuals worldwide. Although its etiopathogenesis remains elusive, increasing evidence shows that genetic susceptibility, environmental factors, intestinal antigens overactivation and aberrant immunological response, as well as intestinal mucosal injury, might contribute to the occurrence and development of UC [1,2]. Therein, the impairment of intestinal barrier function has been considered one of the most important pathophysiologies in UC. Maintaining the integrity of the intestinal barrier and achieving mucosal repair are the goals for the prevention and treatment of UC. Hence, it is essential to extensively explore the potential mechanisms of intestinal barrier dysfunction of UC.

In recent years, emerging evidence has demonstrated the protective role of Vitamin D (VitD) in the development of UC [3]. The VitD level in UC patients is commonly insufficient, and low VitD level is inversely associated with the activity of inflammation [4,5]. On the contrary, VitD supplementation has been illustrated to improve the clinical severity of UC [6]. In addition, results from experimental colitis mice models similarly elucidate that VitD deficiency exacerbates the symptoms of colitis, whereas VitD supplementation alleviates colonic inflammation [7]. Therefore, VitD and its downstream signaling have attracted more and more attention in the aspects of UC prevention and therapy. VitD exerts its versatile bioactivities via binding VitD receptor (VDR). Similarly, the decrease in VDR expression is inversely relative to the activity of UC as well [5,8]. This is affirmed by multiple in vivo studies. Experimental data from *VDR*^−/−^ mice have revealed that VDR deletion could increase gut susceptibility and colitis deterioration responding to DSS insult [9,10]. However, despite the protective role of VitD/VDR in the intestine being well reported [11,12,13], comprehensive underlying mechanisms still necessitate further explorations.

Na^+^/H^+^ exchanger isoform 8 (NHE8, SLC9A8) is a multifunctional transporter involved in sodium uptake, intracellular pH modulation, and cellular volume regulation. Over the last few decades, accumulating evidence has unmasked the essential role of NHE8 in intestinal mucosal protection in infected and inflammatory colitis [14,15]. Our prior study first reported decreased NHE8 expression in UC patients, indicating its potential role in intestinal protection [16]. Moreover, several in vivo and in vitro studies also observed compromised NHE8 expression under inflammatory conditions [15,16,17]. These findings suggest that downregulation of NHE8 is responsible for the development of colitis. Further studies mechanistically revealed that NHE8-mediated intestine protection was involved in colonic mucus secretion [14,18]. In addition, cellular proliferation and intracellular pH control modulated by NHE8 in the colonic epithelium were regarded as important actions as well [19,20]. However, whether NHE8 is involved in VitD/VDR-mediated intestinal protection remains unclear. 

In the current study, we aim to investigate the effects of VitD/VDR on NHE8 regulation under physiological and inflammatory conditions. By using VitD-deficient mice, *VDR*^−/−^mice, *NHE8*^−/−^ mice, Caco-2 cells and intestinal organoids, we perform a series of in vivo and in vitro experiments to elucidate the roles and mechanisms of VitD/VDR in the regulation of NHE8 in colitis. Our findings make novel contributions to better understanding the mechanisms of VitD/VDR in intestinal mucosa protection in UC.

## 2. Materials and Methods

### 2.1. Experimental Animal Models

For the VitD deficiency model, weaned wild-type C57BL/6 mice (3 weeks of age) obtained from Beijing Vital River Laboratory Animal Technology Co., Ltd. (Beijing, China) were either raised on a standard chow (1500 IU/kg cholecalciferol) or VitD-deficient chow (<5 IU/kg cholecalciferol, purified VD deficient diet with normal phosphorus and calcium concentrations). Both types of chow were from Trophic Animal Feed High-Tech Co.,Ltd, China for 6 weeks ad libitum. The *VDR*^−/−^ mice were generated from *VDR^+/−^* mice (The Jackson Laboratories, West Sacramento, CA, USA) and were raised on a high calcium and phosphorus chow containing approximately 2200 IU vitamin D/kg diet, along with 20% lactose, 2% calcium, and 1.25% phosphorus (Trophic Animal Feed High-Tech Co., Ltd, Nantong, China). NHE8 heterozygotes (*NHE8^+/−^*) with C57BL/6 background were established by Bioray Laboratories Inc. (Shanghai, China) using CRISPR/Cas 9 system. The *NHE8*^−/−^ mice were bred from *NHE8^+/−^* mice. The primer sequences for gene-type identification are as follows in Table 1. The gene-type identification was performed according to the indications of Mouse Tail Direct PCR Kit (Bimake, Shanghai, China). All mice in this study were paired with age and sex.

For the colitis model, mice were divided into 6 groups (*WT* + DSS, DSS + Par, VDd + DSS, *VDR*^−/−^ + DSS, *NHE8*^−/−^ + DSS, *NHE8*^−/−^ + DSS + Par) with 8 mice each group and were administrated with 2.5% dextran sodium sulfate (DSS, M.W. 36,000–50,000 Da, S3045, MP Biomedicals, Canada) containing water ad libitum for 7 days, followed by normal drinking water for 3 days. In physiological conditions, mice were divided into 4 groups (CT, VDd, *WT*, *VDR*^−/−^) with 8 mice in each group. For paricalcitol treatment, mice received paricalcitol (300 ng/kg/day) via intraperitoneal injection for 7 days before the first day of drinking DSS-containing water and paricalcitol was administrated during the whole DSS experiment. Disease activity index (DAI) was determined from body weight loss, stool consistency, and fecal hemoccult as described previously [21,22]. All animal experiments were performed following the protocol approved by the Institutional Animal Care and Use Committee of Sichuan University (2017076A, approval date: 17 October 2017) and in compliance with ARRIVE (Animal Research: Reporting of In Vivo Experiments) Guidelines.

### 2.2. Determination of Serum 25(OH)2VitD3, Calcium and Phosphorus Levels

Mice blood samples were collected by removing one of the eyeballs after anesthesia. Samples were kept standing at room temperature for at least 30 min, followed by 3000 rpm centrifugation for 15 min to separate serum and were stored at −80 °C. Serum 25(OH)2VitD3 levels were measured by a commercial Elisa Kit (Cloud-Clone Corp, Houston, TX, USA). Briefly, 1:4 diluted serum samples (3 wells for each sample per experiment) were added into a microwell plate which included pre-coated antibody at 37 °C for 1 h, followed by washing with an auto-cleaning machine 3 times. After incubating with the enzyme-labeled antibody at 37 °C for 1 h, the cleaning solution was used again. Finally, fresh TMB substrate solution was added to the reaction followed by STOP solution. A microplate reader was used to measure the absorbance at 450 nm. Serum calcium and phosphorus levels were detected by an automatic biochemical analyzer (Hitachi, Beijing, China).

### 2.3. Cell Culture and Treatment

Caco-2 cells were cultured in DMEM medium (DMEM, 10313039, Gibco, GrandIsland, NE, USA) with 10% fetal bovine serum (FBS, 1913444, Biological Industries, Jerusalem, Israel) as previously described [16]. Briefly, Caco-2 cells were digested with 0.25% trypsin (J200050, CYTIVA, Washington, DC, USA) for 3 min to make single cells after growth to 60–80% confluence. Suspended cells were added to another sterile cell flask after centrifugation with 1000 rpm for 3 min. All experiments were performed between 4 and 10 passages.

Caco-2 cells were incubated with 100 ng/mL TNF-α (PeproTech, Rocky Hill, CT, USA) for 12 h to mimic inflammatory stimulation. To silence/enhance VDR expression, Caco-2 cells were transfected with human VDR small interfering RNA (siRNA, RiboBio Co., Ltd., Guangzhou, China) or VDR overexpression plasmid (pVDR, HANBIO, Shanghai, China) by using JetPRIME® DNA and siRNA Transfection Reagent (Polyplus-transfection S.A.). The target sequence for VDR was: AGCGCATCATTGCCATACT. For activating VDR, Caco-2 cells were treated with 10 nM paricalcitol for 18 h in the absence or presence of TNF-α. For the mechanical study, NF-κB signaling inhibitor QNZ (100 nM, Selleck, Houston, TX, USA) was exposed to TNF-α-treated Caco-2 cells as previously described [23].

### 2.4. Enteroids Culture

Mouse intestinal stem cells were extracted and cultured using IntestiCultTM organoid growth medium (06005, STEMCELL Technologies) according to the manufacturer’s instructions. Briefly, the 10 cm jejunum adjacent to the stomach was removed and washed with ice-cold D-PBS repeatedly, and then the intestine was cut into 1–2 mm pieces and transferred into tubes with fresh ice-cold D-PBS. The segments were rinsed with ice-cold D-PBS 15–20 times until the supernatant no longer contained visible debris, followed by incubating in Gentle Cell Dissociation Reagent (07174, STEMCELL Technologies) for 20 min at room temperature. Subsequently, the intestine segments were resuspended in PBS supplemented with 0.1% bovine serum albumin. Dissociated intestinal crypts were filtered through 70 mm strainers, resuspended in Dulbecco’s modified Eagle’s medium (DMEM)/F12 medium containing 15 mM HEPES, and counted. The calculated number of dissociated intestinal crypts were resuspended in IntesticultTM organoid growth medium and Matrigel (356230, Corning, New York, NY, USA) in a 1:1 ratio and plated in a 24-well culture plate. After incubating for 10 min, 700 μL complete organoid growth medium was added to each well. The IntesticultTM organoid growth medium was replaced every other day, and the intestinal organoids were cultured and passaged every 7 days.

### 2.5. Western Blot Detection

Total proteins were extracted from mice colon tissues and cultured Caco-2 cells using the RIPA lysis buffer (Beyotime, Shanghai, China) according to the manufacturer’s instructions. The concentration of proteins was detected using BCA Kit (Beyotime, Shanghai, China) and the amount of 50 μg proteins was loaded on the polyacrylamide gels. Briefly, the proteins were separated by SDS-PAGE and transferred to a polyvinylidene fluoride membrane, followed by blocking with 5% non-fat milk. The membranes were incubated with anti-NHE8 (1:1000, Invitrogen, Waltham, CA, USA), anti-VDR (1:1000, Abcam, Cambridge, UK), anti-GAPDH (1:1000, Good Here, Hangzhou, China), anti-phospho-NF-κB p65 (1:1000, CST, Danvers, MA, USA) and anti-NF-κB p65 (1:1000, CST, Danvers, Massachusetts, USA) antibodies at 4 °C overnight. Then the membranes were incubated with appropriate secondary antibodies and detected by an ECL kit (Beyotime, Shanghai, China).

### 2.6. Immunohistochemistry (IHC) and Immunofluorescence (IF) Staining

After being sacrificed by dislocating the cervical vertebrae, mice colonic tissues were fixed in 4% paraformaldehyde and embedded in paraffin, which were cut into 4 µm-thick sections. The sections were deparaffinized and treated with 1×Tris-EDTA for antigen repair. For the IHC assay, sections were treated with 3% hydrogen peroxide to strict endogenous peroxidase activity and blocked with goat serum, followed by incubating with anti-NHE8 (1:200, Invitrogen), anti-VDR (1:100, Abcam) antibodies. Then, goat anti-rabbit secondary antibody, horseradish peroxidase-labeled streptavidin and DAB working solution were used for color development. With regard to IF, colon or enteroid sections were pretreated with 0.2% Triton X-100 for 10 min at room temperature. Subsequently, sections were blocked with goat serum, followed by incubating with anti-NHE8 (1:200, Invitrogen), anti-pp65 (1:100, CST) antibodies and corresponding secondary antibodies (Alexa Fluor-555 or Alexa Fluor-488, Abcam). DAPI staining agent (Sigma, San Francisco, CA, USA) was used for dying cells nuclear. Images were examined using the Zeiss Imager Z2 fluorescence microscope (Carl Zeiss, Berlin, Germany).

### 2.7. Fluorescence In Situ Hybridization (FISH) Experiment

4 µm-thick paraffin-embedded mice colon tissue sections were conventional dewaxed and dehydrated. FISH Kit (#D-0016, FocoFish, Guangzhou, China) was performed according to the manufacturer’s instructions. In short, sections were incubated with Solution A, B and C for 20 min at room temperature, followed by adding Block buffer at 55 °C for 2 h. Then, the EUB388 probe was hybridized with 16sDNA of intestinal bacteria at a 1:50 dilution in a 37 °C incubator for 24 h. Next, a washing buffer was used to clean the residual probe. Finally, sections were dehydrated and dried with gradient alcohol (70–100%), followed by DAPI staining. Images were examined using the Zeiss Imager Z2 fluorescence microscope (Carl Zeiss).

### 2.8. Statistical Analysis

All data were presented as mean ± standard error (SEM). SPSS 21.0 software (SPSS, Chicago, IL, USA) was used to perform statistical analysis. Student’s *t*-test was employed for the two groups’ comparison. A one-way ANOVA test was performed to compare three or more groups. Two-way ANOVA analysis was used to assess between-group differences in factorial design. *p* value less than 0.05 was considered significant. Data in this study were obtained from three independent trials performed in triplicate.

## 3. Results

### 3.1. Paricalcitol Treatment Attenuates DSS-Induced Colitis and Upregulates Colonic NHE8 Expression

We first investigated the protective effects of VitD/VDR in DSS-induced colitis mice. As shown in Figure 1A, DSS challenges obviously increased the DAI scores compared with the control group after the fifth day. Compared with DSS-colitis mice, paricalcitol treatment effectively decreased the DAI scores on the eighth day and prolonged to the tenth day (*p* < 0.05). Meanwhile, paricalcitol treatment partly prevented DSS-induced colon shortening (*p* < 0.05, Figure 1B). Histological assessments by H&E staining and histological scores revealed that paricalcitol treatment prevented DSS-induced mucosal structure damage and decreased the histological scores compared with the DSS-colitis mice (*p* < 0.05, Figure 1C,D). These observations suggested that paricalcitol treatment could ameliorate the colonic mucosal injury under DSS-colitis conditions.

To understand whether paricalcitol-mediated colon protection was dependent on NHE8 regulation, we further detected its expression. As shown in Figure 1E, Western blot analysis indicated that NHE8 and VDR protein expressions were decreased after DSS challenges, whereas paricalcitol treatment prevents the decrease in their levels effectively (*p* < 0.05), respectively. In parallel, IHC staining displayed that paricalcitol treatment partly blocked DSS-induced NHE8 and VDR protein compromises in colonic epithelia, respectively (Figure 1C). Further correlation analysis suggested the expressions of NHE8 and VDR proteins in colitis mice were positively related (*n* = 12, R = 0.653, *p* = 0.021; Figure 1F). Taken together, VitD/VDR-mediated intestinal protection might depend on the stable expression of NHE8.

### 3.2. VitD Deficiency Aggravates DSS-Induced NHE8 Downregulation

To understand whether compromised NHE8 was responsible for the lack of VitD induced, increasing intestinal susceptibility to DSS, a VitD-deficient colitis mice model was further established. As shown in Figure 2A, mice were fed with VitD-deficient chow for 6 weeks and this model was verified by a decrease in serum 25(OH)2 VitD3 without discrepancy in serum calcium and phosphonium (Figure 2B). DSS challenges overtly aggravated the symptoms of VitD-deficient mice since the 3rd day as evidenced by the drastic increase in DAI scores (Figure 2C). Meanwhile, the colon length of VitD-deficient colitis mice was obviously shorter than DSS colitis mice (*p* < 0.05, Figure 2D). Histological examination showed VitD-deficient colitis mice suffered more disastrous damage in the colonic architecture with higher histological scores compared with control colitis mice (*p* < 0.05, Figure 2E,F). 

Moreover, the colonic protein expression of NHE8 in VitD-deficient colitis mice was further detected. IF staining presented an obvious debility of NHE8 in VitD-deficient colitis mice compared with that in control colitis mice (Figure 2E). The DSS-induced NHE8 deprivation in VitD-deficient mice was further verified by Western blot analysis (*p* < 0.05, Figure 2G). In addition, the intestinal barrier function was evaluated by probe EUB338. As shown in Figure 2E, we found that there were more gut bacteria invading into submucosa in VitD-deficient colitis mice, suggesting VitD deficiency weakened the mucosal barrier. These findings demonstrated that NHE8 might participate in VitD deficiency caused by aggravation of the colitis.

### 3.3. NHE8 Expression Is Compromised in VDR^−/−^ Colitis Mice

VitD exerts its gut protective function via binding to VDR. To better understand the regulatory effect of VDR on NHE8 in colitis, we further examined the colonic NHE8 alterations in *VDR*^−/−^mice. Figure 3A depicts the genotype and VDR expression of *VDR*^−/−^ mice. Owing to the high calcium and phosphorus chow feeding, the levels of serum calcium and phosphorus were comparable between *WT* and *VDR^−/−^
*(Figure 3B). In accordance with VitD-deficient mice, *VDR*^−/−^ colitis mice exhibited higher DAI scores (Figure 3C), shorter colon length (Figure 3D), and more serious colonic structure deterioration compared with *WT* colitis mice (Figure 3E,F). Western blot and IF staining, meanwhile, revealed evident NHE8 deprivation in *VDR*^−/−^ colitis mice compared with *WT* colitis mice (*p* < 0.05, Figure 3E,G). Similarly, *VDR*^−/−^ colitis mice displayed more enhanced permeability of luminal bacteria from colonic apical membrane to lamina propria and more weakened barrier function compared with *WT* colitis mice (Figure 3E). These observations suggested that the downregulation of NHE8 proteins might be involved in VDR-mediated intestinal protection in colitis.

### 3.4. Paricalcitol Failed to Restore DSS-Induced Intestinal Injury in NHE8^−/−^ Mice 

To deeply investigate the pivotal role of NHE8 in VitD/VDR-mediated intestine protection, *NHE8*^−/−^ mice were further employed. The genotype and NHE8 expression of this mice model were verified in Figure 4A,B. Then, the effects of paricalcitol on DSS-induced *NHE8*^−/−^ colitis mice were examined. Compared with *WT* colitis mice, the DAI scores in *NHE8*^−/−^ colitis mice obviously increased after the 4th day and prolonged to the end of the experiment, whereas paricalcitol treatment failed to alleviate the DAI scores (Figure 4C). The histological observations also demonstrated more serious colonic mucosal injury in *NHE8*^−/−^ colitis than that in *WT* colitis mice, whereas paricalcitol could not restore the mucosal damage (Figure 4D,E). EUB338 probe assay also exhibited more colonic bacteria translocated from the intestinal lumen to the submucosa in *NHE8*^−/−^ colitis mice, and paricalcitol could not reconstruct normal barrier function (Figure 4D). These findings corroborated that VitD/VDR-mediated intestine protection was dependent on NHE8 regulation. 

### 3.5. Colonic NHE8 Expression Is Modulated by VitD/VDR under Physiological Condition

As VitD deficiency and VDR depletion were inversely related to the activity of colitis which was attributed to the downregulation of NHE8 expression, we wondered whether VitD absence/VDR knockdown leading to intestinal barrier dysfunction is implied with NHE8 downregulation under physiological conditions. We first examined NHE8 expression by Western blot and IF assays. As shown in Figure 5A,B, VitD-deficient chow feeding significantly downregulated colonic NHE8 proteins compared with the control mice (*p* < 0.001). Similarly, the EUB338 probe assay exhibited more invasion bacteria in VitD-deficient mice than in control mice (Figure 5A). Furthermore, NHE8 downregulation resulting from VitD deficiency was obviously restored by paricalcitol treatment (*p* < 0.05, Figure 5C). 

We next determined NHE8 expression in *VDR*^−/−^ mice. As shown in Figure 5D,E, genetic depletion of VDR apparently weakened the fluorescence intensity of NHE8 on colonic apical membrane in *VDR*^−/−^ mice. Western blot assays also verified NHE8 downregulation due to VDR deletion (*p* < 0.001, Figure 5F). Moreover, intestinal function assay also showed obvious gut flora invasion in *VDR*^−/−^ mice (Figure 5D). In general, we drew the conclusion that NHE8 was involved in VitD/VDR-mediated intestine protection under physiological conditions. 

### 3.6. VitD/VDR Regulates NHE8 Expression in Caco-2 Cells 

As the Caco-2 cell line was widely used in intestinal research, we further examined the regulatory effects of VitD/VDR on NHE8 in vitro. As shown in Figure 6A,B, we found NHE8 protein expression was significantly decreased under inflammatory conditions induced by TNF-α exposure compared with the control group (*p* < 0.01), whereas paricalcitol treatment obviously restored NHE8 proteins (*p* < 0.05). Meanwhile, IF staining also revealed paricalcitol partly blocked TNF-α-induced a decrease in NHE8. 

Then, to manipulate the regulatory effects of VDR on NHE8, we used specific VDR siRNA and overexpression plasmid to intervene in Caco-2 cells. As shown in Figure 6C,D, the transfected efficiency was affirmed by Western blot detection. VDR siRNA could decrease VDR protein level to approximately 50% (*p* < 0.001), and overexpression plasmid enhanced its expression more than 1.2 times than that of the vehicle group (*p* < 0.01). Silencing VDR markedly downregulated NHE8 proteins compared with vehicle-treated cells (*p* < 0.01), however, pVDR transfection obviously upregulated NHE8 level (*p* < 0.05). Taken together, these findings further illuminated the regulatory role of VitD/VDR in NHE8 expression.

### 3.7. VDR Upregulates NHE8 Expression through Suppressing NF-κb p65 Signaling Pathway in Colitis 

NF-κb p65 signaling activation has been suggested as a prominent mechanism involved in the development of colitis, and previous studies have demonstrated the regulatory role of VDR in NF-κb p65 signaling [24]. Therefore, we wanted to explore whether VDR-mediated NHE8 regulation was dependent on NF-κb p65 signaling pathway in colitis. We first examined the activation of NF-κb p65 signaling in VitD-deficient and *VDR*^−/−^ colitis mice by detecting the nuclear location of pp65. As shown in Figure 7A, IF staining revealed DSS treatment potentiated fluorescence intensity of pp65 in cellular nuclear, indicating the activation of NF-κb p65 signaling. Furthermore, stronger pp65 fluorescence was observed in VitD-deficient colitis mice. Similarly, consistent changing trends of nuclear location and fluorescence intensity of pp65 were observed in *VDR*^−/−^ colitis mice (Figure 7B).

The regulatory effects of VDR to p65 phosphorylation were further determined in Caco-2 cells. Results from Figure 7C revealed that TNF-α exposure significantly stimulated the levels of phosphorylated NF-κb p65 compared with the control cells (*p* < 0.001). Notably, VDR siRNA pre-transfection dramatically augmented TNF-α-stimulated NF-κb p65 phosphorylation compared with vehicle-TNF-α treated cells (*p* < 0.001), whereas the stimuli were effectively blocked by NF-κb p65 inhibitor QNZ intervention. These findings demonstrated that VDR downregulation synergistically promoted inflammation-triggered activation of NF-κb p65 signaling.

Based on the above observations, we further employed QNZ to determine NHE8 alterations. As shown in Figure 7D, QNZ exposure effectively blocked TNF-α-induced NHE8 decrease (*p* < 0.05). Meanwhile, VDR siRNA transfection dramatically aggravated the TNF-α-induced decrease in NHE8 expression (*p* < 0.05), whereas this downregulation was effectively blocked in the presence of QNZ (*p* < 0.01). These findings suggested that VDR regulated NHE8 expression by suppressing NF-κb p65 activation under inflammatory conditions. 

Finally, we exploited the 3D-organoid system to fully demonstrate the mechanism of VDR-mediated NHE8 expression. Figure 7E displays the morphological alternations of intestinal crypts and we detected protrusions appearing at the periphery of enteroids, denoting that intestinal organoids gradually mature over time. The results from IF staining showed that the fluorescence intensity of NHE8 was obviously reduced in *VDR*^−/−^ enteroids compared with wild-type enteroids, whereas the fluorescence intensity of nuclear pp65 was obviously strengthened (Figure 7F). Above all, these consequences further supported the regulatory effects of VitD/VDR on the NHE8 and NF-κb p65 signaling pathway.

## 4. Discussion

VitD/VDR has been suggested to play a pivotal role in intestine protection; however, the potential molecular mechanism remains poorly understood. In the current study, we uncovered a novel mechanism by which VitD/VDR prevented intestinal barrier dysfunction by stimulating NHE8 expression in colitis, highlighting its promising role in repairing intestinal mucosal injury. 

Multiple clinical studies have reported that IBD patients are more prone to exert VitD deficiency than healthy people, and VitD level is negatively associated with the activity of IBD, indicating the potential protective role of VitD and its downstream signaling in IBD [25,26]. Thus, exploring the mechanisms of VitD deficiency in the development of colitis could provide explicit therapeutic targets for IBD. At present, a series of attempts have been made. Although current data have indicated the potential role of VitD/VDR in the regulation of the immune system, gut microbiota and tight junction barrier [12,27,28], the potential mechanisms of VitD/VDR have necessitated further excavations. Over the past decade, ours and other research groups reported a novel Na^+^/H^+^ exchanger NHE8 participating in the protection of the intestinal barrier [14,15,16,17,18,20]. As a result, we aimed to explore whether NHE8 was involved in VitD/VDR-mediated intestinal protection. To address our speculations, we first estimated the effects of paricalcitol (vitamin D receptor activator) on NHE8 in colitis mice. Consistent with previous studies, paricalcitol treatment partly alleviated DSS-induced symptoms and colonic mucosal structure injury [29,30]. Simultaneously, along with the mucosal restoration, DSS-challenge-caused colonic NHE8 insults were significantly improved by paricalcitol treatment, suggesting that NHE8 was implicated in VitD-mediated intestine protection. Furthermore, as other investigators and our data revealed that VitD deficiency remarkedly aggravated DSS-induced colitis manifesting higher DAI scores and histological scores [7,31], we wanted to know whether NHE8 was involved in colitis aggravation. Notably, except for the aggravation of colitis, dramatical NHE8 deprivation in VitD deficiency colitis mice was also observed, indicating that the downregulation of NHE8 caused by VitD deficiency was implicated in the aggravation of colitis. The role of NHE8 in VitD-deficient colitis was further verified in *NHE8*^−/−^ colitis mice. In mice challenged by DSS, genetic NHE8 deletion obviously exacerbated intestinal impairment and bacteria translocation, whereas paricalcitol failed to alleviate symptoms and restore colonic histological structure injury in *NHE8*^−/−^ mice challenged with DSS. Therefore, these findings further indicated the indispensable role of NHE8 in VDR-mediated intestinal protection. 

Meanwhile, well-known as the predominant receptor mediating most functions of VitD and an IBD risk gene, increasing evidence from clinical and experimental studies has illuminated the protective role of VDR in intestine protection [10,32,33]. Currently, the available studies suggest VDR executes intestinal protection by regulating multiple patterns such as intestinal proliferation and differentiation, intestinal barrier function, intestinal immunity, microbial homeostasis and so on [22,34]. However, whether NHE8 was involved in VDR-mediated intestinal protection remains unclear. As a result, based on the observations of NHE8 alterations in VitD-deficient mice, we were interested in exploring the regulatory role of VDR in NHE8 expression. First, in line with a previous study [35], genetic deletion of VDR markedly increased the susceptibility of DSS as evidenced by exacerbating colonic mucosal injury, higher DAI scores and increased bacteria translocation. Furthermore, except for colitis deterioration, VDR deletion also caused a dramatic reduction in NHE8 expression compared with wild-type colitis mice. These findings substantiated that VDR was indispensable in the gut epithelium, and loss of VDR could increase intestinal susceptibility to inflammation by suppressing NHE8 expression. 

In addition, to deeply understand whether NHE8 was responsible for VitD-deficiency-related colitis, we further investigated NHE8 levels in VitD-deficient mice in healthy states. Unexpectedly, in VitD-deficient mice and *VDR*^−/−^ mice, the intestinal function was impaired evidenced by increased bacteria translocation, and this further illuminated the protective role of VitD/VDR in intestinal barrier function. In addition, the expression of colonic NHE8 in VitD-deficient mice was significantly downregulated, whereas supplementation of paricalcitol effectively restored NHE8 expression. Consistent with VitD-deficient mice, the expression of NHE8 in *VDR*^−/−^ mice was significantly downregulated. These observations not only confirmed the positive role of VitD/VDR in NHE8 regulation, but also implied that the decrease in NHE8 expression under healthy states was a crucial predisposing factor promoting the development of colitis. In fact, genetic deletion of NHE8 could trigger a colonic inflammatory response and upregulate the expression of inflammatory cytokines, which further resulted in the gut hyper-susceptibility to DSS [14]. However, although VitD-deficiency-induced NHE8 reduction was responsible for the development of UC remained elusive, our findings provided strong evidence that NHE8 downregulation was responsible for VitD-deficiency-caused colitis aggravation.

As a series of studies revealed the protective functions of NHE8 on epithelium through regulating cellular proliferation and pHi [16,19,20], we proceeded to investigate the effects of VitD/VDR on NHE8 regulation in Caco-2 cells and enteroids. Under inflammatory conditions, the decrease in NHE8 was subject to TNF-α signaling [17,36]. Therefore, we established an in vitro inflammatory model by TNF-α exposure. Consistent with the observations of in vivo studies, paricalcitol exposure effectively blocked TNF-α-induced NHE8 downregulation, whereas VDR silence compromised NHE8 expression and further sharply promoted TNF-α-induced NHE8 loss. On the other hand, inhibition of intestinal NHE8 expression could stimulate colitis-activated TNF-α expression, indicating the interplay of NHE8 and TNF-α [14]. As NF-κb p65 was a paramount pathway monitoring intestinal inflammation in response to TNF-α [37], and VDR has been reported to regulate NF-κb activity negatively in the intestine [24], we therefore detected the role of NF-κb p65 signaling in VitD/VDR-mediated NHE8 regulation. Regarding the results from in vivo studies, we observed obvious active NF-κb p65 signaling in VitD-deficient and *VDR*^−/−^ colitis mice compared to wild-type colitis mice, indicating that the suppression of the VitD/VDR pathway stimulated the overactivation of NF-κb p65 signaling in the challenge of DSS. This result was further affirmed in Caco-2 cells treated with VDR siRNA and TNF-α. In fact, even under physiological conditions, suppression of VDR was observed to stimulate NF-κb p65 signaling pathway in the colonic epithelium [24] and was in accordance with our 3D-organoid experiment. Our experiment data also indicated another inflammatory response signaling: ERK1/2 MAPK signaling was activated in VitD deficiency mice. These findings demonstrated the positive role of VitD/VDR in inflammation regulation. As NHE8 was subjected to inflammation, to explore the role of NF-κb p65 signaling in VDR-mediated NHE8 regulation, the specific inhibitor QNZ was employed. Noteworthily, the deprived NHE8 expression induced by TNF-α or TNF-α-combining VDR siRNA was effectively antagonized by QNZ exposure. These observations provided direct evidence that VDR regulated NHE8 expression by modulating NF-κb p65 signaling pathway to participate in intestinal barrier protection. However, the potential modulatory mechanisms of which NF-κb p65 regulated NHE8 warrant further investigation. 

## 5. Conclusions

In summary, our in vivo and in vitro data recapitulated that depressed NHE8 expression was responsible for VitD deficiency and VDR suppression-induced colitis aggravation. We further revealed the important role of the NF-κb p65 signaling pathway in VitD/VDR-mediated NHE8 regulation. In conclusion, our results make novel improvements in understanding the molecular mechanisms, that is, VitD/VDR protects against intestinal inflammation in UC, and provide strong evidence supporting the viewpoint that maintaining sufficient VitD concentration is important for IBD patients.

## 6. Limitation

The restrictions in our study are that we only assessed the changes in NHE8 protein expression under colitis and physiological conditions and we did not detect mRNA expression. Another shortcoming is that we did not evaluate whether VDR binds to NHE8 promoters to activate its function and only determined one mechanism for VDR upregulating NHE8 in enteroids. Further in-depth studies are required to understand the molecular mechanisms between VDR and NHE8.

## Figures and Tables

**Figure 1 nutrients-15-04834-f001:**
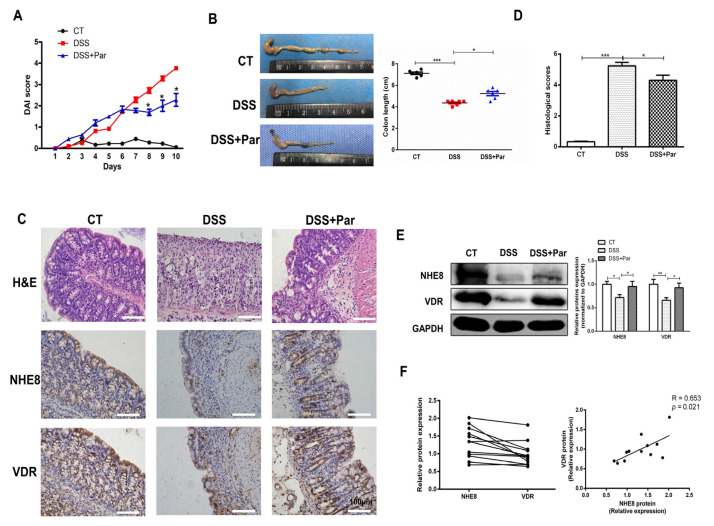
Paricalcitol treatment attenuates DSS-induced colitis and upregulates colonic NHE8 expression. (**A**): Disease activity index (DAI) scores of mice during the experiment. (**B**): Representative macrophotograph of colon (left) and statistical analysis of colon length in each group (right). Red represents DSS group and Blue represents DSS + Par group (**C**): Representative hematoxylin and eosin (H&E) staining (top), and immunohistochemical staining for NHE8 (middle) and VDR (bottom) of colon tissues from mice in each group. (**D**): Statistical analysis of histological scores for colon. (**E**): Western blot detection of colonic NHE8 and VDR protein expression in each group. (**F**): Correlation analysis for colonic NHE8 and VDR proteins in DSS-treated colitis mice. CT: control group mice; DSS: DSS-induced colitis mice; DSS + Par: DSS-induced colitis mice with paricalcitol treatment; *n* = 6 in each group. * *p* < 0.05, ** *p* < 0.01, *** *p* < 0.001 for statistical significance.

**Figure 2 nutrients-15-04834-f002:**
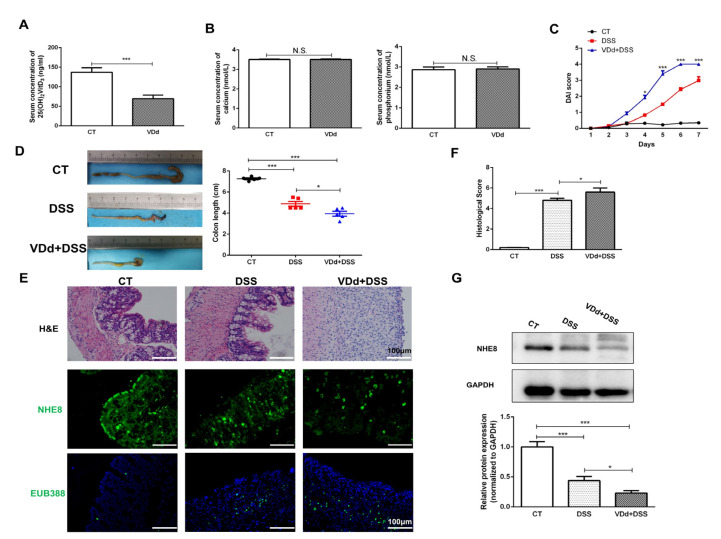
VitD deficiency aggravates DSS-induced NHE8 downregulation. (**A**): Serum 25(OH)2VitD3 levels were measured by Elisa assay. (**B**): Serum calcium and phosphorus concentration were detected by biochemical assay. (**C**): DAI scores of control mice (CT), DSS-induced colitis (DSS) mice and VitD deficient (VDd) colitis mice. (**D**): Colon length of CT, DSS and VDd mice. Red represents DSS group and Blue represents VDd+DSS group (**E**): H&E (upper), NHE8 (middle) and EUB388 (lower) staining for mice colon tissues. (**F**): Statistical analysis of histological scores for colon. (**G**): Western blot detection of colonic NHE8 protein expression in each group. CT: control mice; VDd: VitD deficient mice; DSS: DSS-induced colitis mice; VDd + DSS: VitD deficient mice with DSS-induced colitis. * *p* < 0.05, *** *p* < 0.001, N.S.: not significant.

**Figure 3 nutrients-15-04834-f003:**
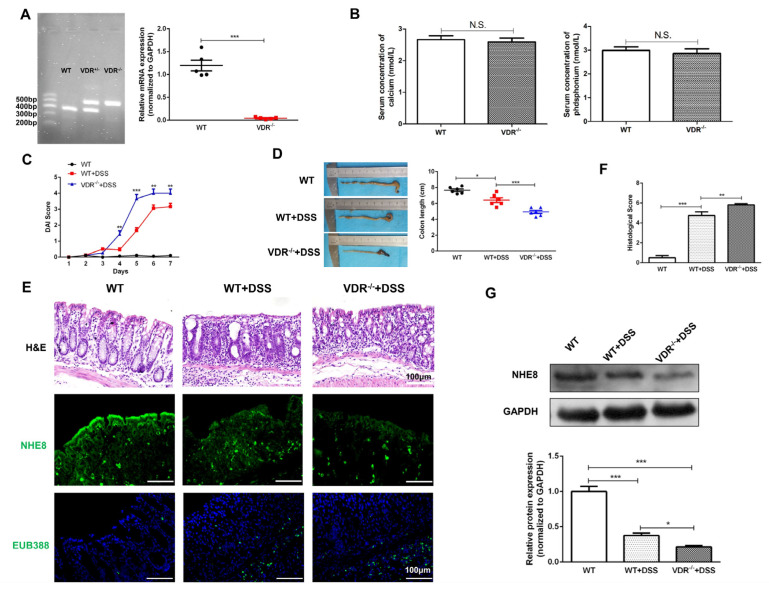
NHE8 expression is compromised in *VDR*^−/−^ colitis mice. (**A**): Genotype identification of *VDR*^−/−^ mice by agarose gel (left), and VDR mRNA detection by qPCR (right). (**B**): Serum calcium and phosphorus concentration of wild-type mice (*WT*) and VDR knockdown (*VDR*^−/−^) mice. (**C**): DAI assessment of each group. (**D**): Representative macrophotograph of colon (left) and statistical analysis of colon length in each group (right). Red represents WT+DSS group and Blue represents *VDR*^−/−^+DSS group (**E**): H&E (upper), NHE8 (middle) and EUB388 (lower) staining in each group. (**F**): Statistical analysis of histological scores for colon. (**G**): Western blot detection of colonic NHE8 protein expression in each group. * *p* < 0.05, ** *p* < 0.01, *** *p* < 0.001 for statistical significance.

**Figure 4 nutrients-15-04834-f004:**
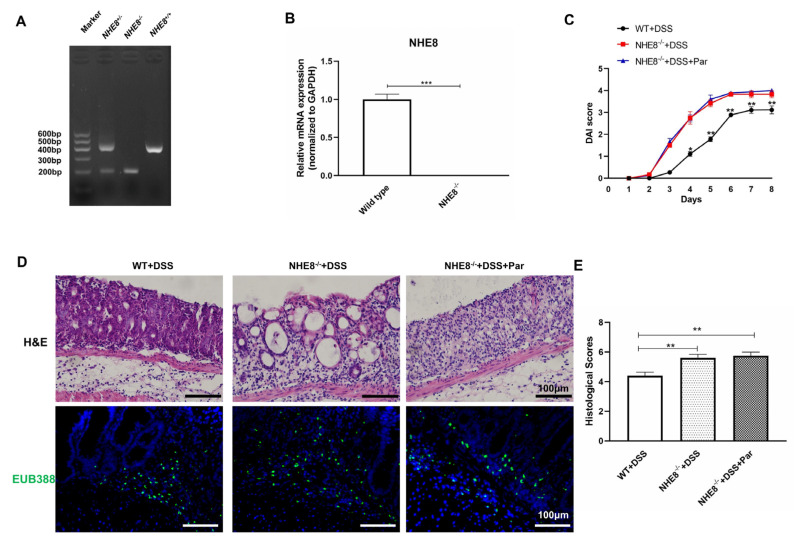
Paricalcitol failed to restore DSS-induced intestinal injury in *NHE8*^−/−^ mice. (**A**): Genotype identification of NHE8 knockdown (*NHE8*^−/−^) mice. (**B**): NHE8 mRNA detection by qPCR. (**C**): DAI scores in each group mice during the experiment. (**D**): H&E (upper) and EUB388 (lower) staining in each group. (**E**): Statistical analysis of histological scores for colon tissues. * *p* < 0.05, ** *p* < 0.01, *** *p* < 0.001 for statistical significance.

**Figure 5 nutrients-15-04834-f005:**
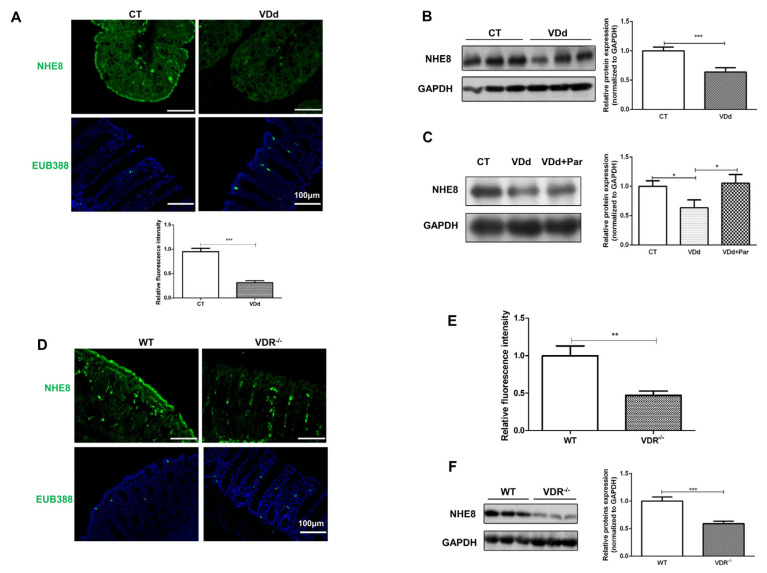
Colonic NHE8 expression is modulated by VitD/VDR under physiological conditions. (**A**): NHE8 (upper) and EUB388 (lower) staining in VitD deficient mice under physiological conditions. (**B**): Western blot detection of colonic NHE8 protein expression in VitD deficient mice. (**C**): Paricalcitol treatment recovered compromised NHE8 expression. (**D**): NHE8 (upper) and EUB388 (lower) staining in *VDR*^−/−^ mice under physiological condition. (**E**): Quantitative description of NHE8 fluorescence intensity. (**F**): Western blot detection of colonic NHE8 protein expression in *VDR*^−/−^ mice. * *p* < 0.05, ** *p* < 0.01, *** *p* < 0.001 for statistical significance.

**Figure 6 nutrients-15-04834-f006:**
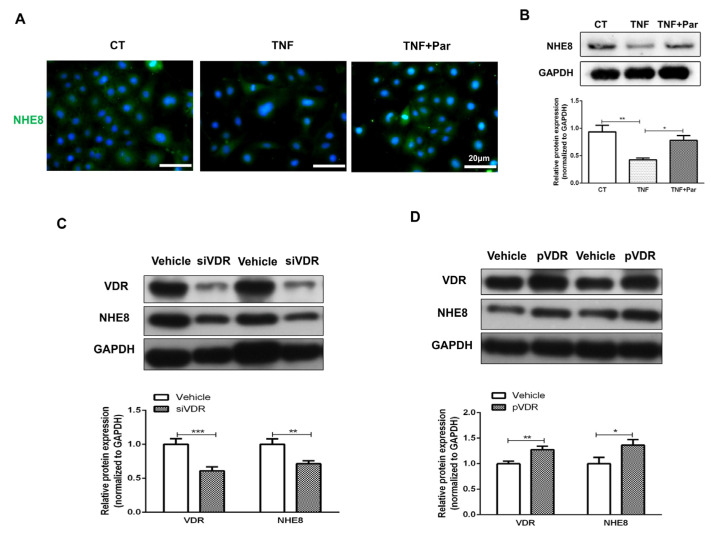
VitD/VDR regulates NHE8 expression in Caco-2 cells. (**A**): Representative immunofluorescence staining for NHE8 protein in Caco-2 cells (green: NHE8; blue: cell nucleus). (**B**): Western blot detection of NHE8 expression in Caco-2 cells. (**C**): Western blot detection of NHE8 and VDR expression in VDR siRNA transfected Caco-2 cells. (**D**): Western blot detection of NHE8 and VDR expression in VDR overexpression plasmid transfected Caco-2 cells. CT: control cells; TNF: TNF-α treated cells; TNF + Par: paricalcitol and TNF-α co-treated cells; siVDR: VDR siRNA transfected cells; pVDR: VDR overexpression plasmid transfected cells. * *p* < 0.05, ** *p* < 0.01, *** *p* < 0.001 for statistical significance.

**Figure 7 nutrients-15-04834-f007:**
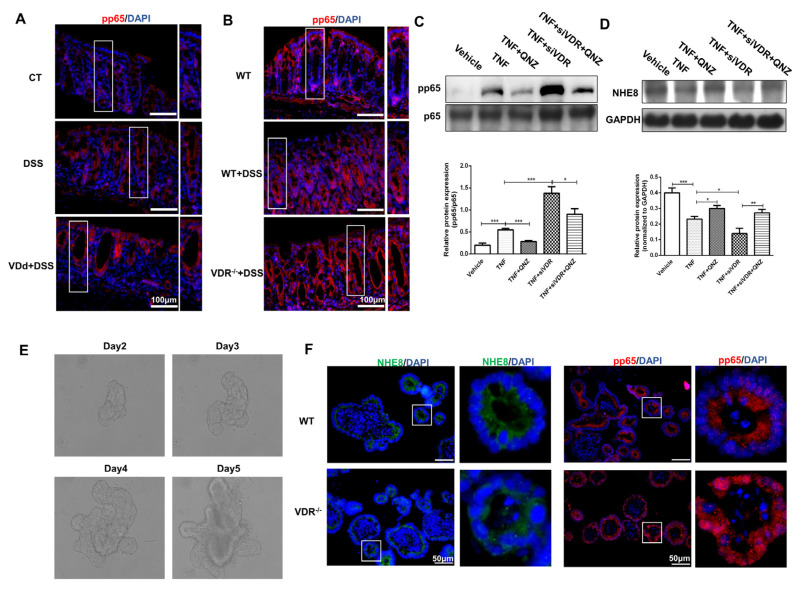
VDR upregulates NHE8 expression by suppressing NF-κb p65 signaling pathway in colitis. The white rectangles show the enlarged part. (**A**): Representative immunofluorescence staining for colonic pp65 proteins in control mice (CT), VitD-sufficient mice with DSS-induced colitis (DSS) and VitD-deficient colitis mice (VDd + DSS). (**B**): Representative immunofluorescence staining for pp65 proteins in wild-type mice (*WT*), wild-type mice with DSS-induced colitis (*WT* + DSS) and *VDR*^−/−^ mice with DSS-induced colitis mice (*VDR*^−/−^ + DSS). Red: pp65; blue: cell nucleus. (**C**): Western blot detection of phosphorylated NF-κb p65 (pp65) and total NF-κb p65 (p65) proteins in Caco-2 cells. (**D**): Western blot detection of NHE8 protein expression of Caco-2 cells in each group. (**E**): Intestinal enteroids extraction and culture in vitro. (**F**): Representative immunofluorescence staining for NHE8 (green) and pp65 (red) proteins in 3D-organoids of *WT* and *VDR*^−/−^ mice. Vehicle: control cells; TNF: vehicle and TNF-α treated cells; TNF + QNZ: QNZ and TNF-α co-treated cells; TNF + siVDR: VDR siRNA transfected and TNF-α treated cells; TNF + siVDR + QNZ: QNZ and TNF-α co-treated cells with VDR siRNA pre-transfection. WT: wild-type enteroids. *VDR*^−/−^: *VDR*^−/−^ enteroids. * *p* < 0.05, ** *p* < 0.01, *** *p* < 0.001 for statistical significance.

**Table 1 nutrients-15-04834-t001:** Mice gene-type identification.

Gene Name	Primer Sequences
*VDR*	Forward: TTCTTCAGTGGCCAGCTCTT
	Mutant-Reverse: CACGAGACTAGTGAGACGTG
	Wild-type-Reverse: CTCCATCCCCATGTGTCTTT
*NHE8*	Forward: TGGGTGGATGACTGTAGTTT
	Mutant-Reverse: TCATGGGTTGTGTTGGGAGA
	Wild-type-Reverse: CTAGGCCTGACGTCAGACC

## Data Availability

The data presented in this study are available on request from the corresponding author.

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
