# Peer review of "Compromised NHE8 Expression Is Responsible for Vitamin D-Deficiency Induced Intestinal Barrier Dysfunction"

_nutrients, 2023, doi:10.3390/nu15224834_

Round 1
Reviewer 1 Report
Comments and Suggestions for Authors
In their manuscript, Guo. Y et al investigate the link between the protective role of vitamin D in colitis and the expression of NHE8. Using an experimental mouse model of chemically-induced colitis, an immortalized colonic cell line and mouse organoids, the authors proposed that NHE8 is regulated by vitamin D, and that the expression of NHE8 is required for the preventive role of vitamin D. These results contribute to a better understanding of the mechanism underlying the effects of vitamin D in colitis, but to reach these conclusions, several major points need to be addressed.
Major comments.
- In the material and methods section, the authors stated that paricalcitol was administered 7 days prior the DSS. Was it also administered after? If not, the author should be careful with their statement as they have determined the protective effect of paricalcitol and not its therapeutic potency. For instance, line 195, the statement ‘paricalcitol treatment effectively restored their levels’ is not correct, as it is the pre-treatment that prevents the decrease of their expression.
- According to this statement, paricalcitol has an effect only when the DSS is removed. Could the authors comment on this observation?
- What was use as a control group for the in vivo experiments in Figure 1? To support the various claims of the authors, several controls are missing: Figure 1, a cohort pre-treated with paricalcitol alone ; Figure 2, a cohort of VitD-deficient mice ; Figure 3, a cohort of VDR-null mice fed with a high calcium and phosphate diet ; Figure 4, a cohort of NHE8-null mice and WT+DSS+Par ; Figure 6, paricalcitol alone ; Figure7A, VitD-deficient and VDR-null cohorts.
- To ensure that NHE8 and VDR colocalize, please provide co-staining experiments.
- Is NHE8 expressed in the epithelium (villin positive cells), crypt (LGR5 positive cells) or both?
- What are the VDR levels in NHE8 null mice.
- What are the VDR protein levels in caco2 cells exposed with TNF?
- For VDR and NHE8 stainings, the representative pictures for the DSS groups are different than the H&E stainings. Few epithelial cells are present. Please show a more representative picture.
- As NHE8 plays a critical role in mucosal protection, please add the histology and the DAI score for the NHE8-null mice.
- What about the expressions of tight junction proteins (e.g. ZO-1) in the various experimental settings?
- NHE8 immunoblottings (Fig. 6B, 7D) and stainings (Fig 6A, 7F) indicate that NHE8 is not strongly expressed in caco2 cells and in organoids derived from WT crypts. Please provide immunoblottings and IF experiments on NHE8-null cells to confirm the signal over the background.
- Is there any putative vitamin D response element in the promoter region of NHE8?
Minor comments :
- Please provide the calcium, vitamin D and phosphate contents in the various diets used in the study.
- Please provide details on the biochemical assays used for calcium and phosphate measurements.
- What was the amount of proteins loaded on the acrylamide gels?
- What were the post-hoc tests used for the one- and two-way ANOVA?
- What does it mean ‘a kind of vitamin D analogs’ (line 403).
Author Response
Dear reviewer,
Re: Manuscript Number: nutrients-2684807 and Title: Compromised NHE8 Expression Is Responsible for Vitamin D-deficiency Induced Intestinal Barrier Dysfunction.
Thank you for your comments concerning our manuscript. Those comments are valuable and very helpful. We have read through comments carefully and have made corrections. Revisions in the text are shown using red highlight for additions. The responses to your comments are presented following. Please see the attachment.
We highly appreciate your time and consideration.
Sincerely,
Corresponding author:
Chunhui Wang
E-mail: wangch@scu.edu.cn
Xiao Li
E-mail: lixiao5166@hotmail.com
Reviewer 1
Major comments.
Q1. In the material and methods section, the authors stated that paricalcitol was administered 7 days prior the DSS. Was it also administered after? If not, the author should be careful with their statement as they have determined the protective effect of paricalcitol and not its therapeutic potency. For instance, line 195, the statement ‘paricalcitol treatment effectively restored their levels’ is not correct, as it is the pre-treatment that prevents the decrease of their expression.
Response:In our study, paricalcitol was pre-treated to mice for 7 days before DSS, inducing the up-regulated VDR expression and it was continued to be administered during the animal experiment. According to your better advice, we have revised the statement at line 101 and 211 marked in red highlight as “paricalcitol treatment prevents the decrease of their levels effectively.”
Q2. According to this statement, paricalcitol has an effect only when the DSS is removed. Could the authors comment on this observation?
Response:Many studies and our previous work have both showed that the colonic inflammation still aggravated continuously after DSS withdraw. The colitis did not stop although mice drunk normal water for the following 3 days [1]. Therefore, our data in this study indicated that paricalcitol had a full-course and persistent effect on colon protection during the DSS-drinking water only differences be observed after 7 days of DSS.
Q3. What was use as a control group for the in vivo experiments in Figure 1? To support the various claims of the authors, several controls are missing: Figure 1, a cohort pre-treated with paricalcitol alone; Figure 2, a cohort of VitD-deficient mice; Figure 3, a cohort of VDR-null mice fed with a high calcium and phosphate diet; Figure 4, a cohort of NHE8-null mice and WT+DSS+Par; Figure 6, paricalcitol alone; Figure7A, VitD-deficient and VDR-null cohorts.
Response:
In the first three parts, we explored the protective role of VitD/VDR/NHE8 mainly in colitis condition, therefore we do not build paricalcitol-treated cohort in Figure 1, VitD-deficient cohort in Figure 2, VDR-null cohort in Figure 3 under physiological status. To further determine the effects of VitD/VDR on NHE8 expression, we treated Caco-2 cells with paricalcitol and found the expression of NHE8 was evaluated (pictures below). Meanwhile, in Figure 6D, NHE8 level was increased after VDR over-expression and in Figure 5C, paricalcitol treatment prevents the decrease of NHE8 in VitD-deficient condition, demonstrating that VitD/VDR/paricalcitol could regulate the expression of NHE8 under physiological condition.
According to Figure 4, we performed FISH and H&E staining for WT/WT+DSS/NHE8-/-/ NHE8-/-+DSS/WT+DSS+Par/ NHE8-/-+DSS+Par groups (picture blow).
According to Figure 6, we treated Caco-2 cells with paricalcitol and found the expression of NHE8 was evaluated (pictures below).
According to Figure 7, as we wanted to explore the regulative effects of VDR on NHE8 in colitis mainly, we did not bulid VitD-deficient and VDR-null cohorts in physiological status. The expression of NHE8 and pp65 in VDR-null cohorts under physiological status was showed in Figure7F in enteroids.
Q4. To ensure that NHE8 and VDR colocalize, please provide co-staining experiments.
Response:The predominant location of NHE8 is on cytomembrane and organelle membrane, but VDR exists mainly in cytoplasm and exerts its function in nucleus. Therefore, we speculated that there was hardly combination of VDR and NHE8 on spatial orientation, and we did not perform the co-staining experiments. If reviewer consider that it is essential, we will carry out it as soon as possible.
Q5. Is NHE8 expressed in the epithelium (villin positive cells), crypt (LGR5 positive cells) or both?
Response:NHE8 is a kind of trans-membrane protein, mainly localized on epithelium (our published article [2]). In recent years, other researchers have shown that NHE8 also localizes on the membrane of LGR5+ stem cells [3].
Q6. What are the VDR levels in NHE8 null mice.
Response:In our previous study, we detected the level of VDR in NHE8 null mice colon and found there was no difference between WT and NHE8-/- mice (pictures below).
Q7. What are the VDR protein levels in caco2 cells exposed with TNF?
Response:In our previous study, we used TNFα to treat Caco-2 cells and detected the expression of VDR. The VDR protein levels were decreased when exposed with TNF and they were restored when paricalcitol treated (pictures below).
Q8. For VDR and NHE8 stainings, the representative pictures for the DSS groups are different than the H&E stainings. Few epithelial cells are present. Please show a more representative picture.
Response:We have replaced more representative picture of H&E staining in Figure 1C.
Q9. As NHE8 plays a critical role in mucosal protection, please add the histology and the DAI score for the NHE8-null mice.
Response:DAI scores are used to evaluate the severity of colitis, so we do not assess it under normal conditions in WT and NHE8-null mice. Consistent with our outcomes, other researches both showed that NHE8-null mice displayed higher DAI scores that WT mice during DSS-drinking water [4]. For histological scores, we found there was no significant difference between WT and NHE8-null mice, but our partial data (below) and other researchers’ studies [4] both showed that there were more bacteria adhesion to colon epithelium in NHE8 knockout mice under normal conditions, indicating that NHE8 plays a pivotal role in intestinal protection.
Q10. What about the expressions of tight junction proteins (e.g ZO-1) in the various experimental settings?
Response:In our published study [5], we found ZO-1 and Claudin-1 were down-regulated in colitis mice colon and the depletion of VDR could aggravate the decrease of tight junctions in colonic inflammation. Under physiological conditions, the expressions of tight junctions were also regulated by VDR.
Q11. NHE8 immunoblottings (Fig. 6B, 7D) and stainings (Fig 6A, 7F) indicate that NHE8 is not strongly expressed in caco2 cells and in organoids derived from WT crypts. Please provide immunoblottings and IF experiments on NHE8-null cells to confirm the signal over the background.
Response:According to your advice, we have detected the level of NHE8 in WT and NHE8-null enteroids to confirm the signal over the background (picture below) and found that there was little fluorescent intensity of NHE8 in NHE8-/- organoid than that in WT organoid. Meanwhile, in our published study, we detected the NHE8 location and expression, confirming the high expression of NHE8 in human colon epithelium adequately.
Our published data: NHE8 expression in human colorectal mucosa in controls (CT) and UC patients (UC)
Q12. Is there any putative vitamin D response element in the promoter region of NHE8?
Response:We used JASPAR software to forecast the combination region of vitamin D receptor response element to NHE8 promoter and found there were putative vitamin D response element in the promoter region of NHE8 (below) in Homo species/ Mus species. However, it is limitation of our study that the double luciferase reporting experiment is needed to carry out and we will pay attention to it in our future experiments.
Human Jaspar VDR-NHE8 Mouse Jaspar VDR-NHE8
Minor comments:
Q1. Please provide the calcium, vitamin D and phosphate contents in the various diets used in the study.
Response:We have added the main ingredient of our various diets in “2.1. Experimental animal models” using red highlight for additions at line 81-83, 86-87.
Q2. Please provide details on the biochemical assays used for calcium and phosphate measurements.
Response:The serum calcium and phosphorus levels were detected by an automatic biochemical an-alyzer (Hitachi, China). We have added detailed information in “2.2. Determination of serum 25(OH)2VitD3, calcium and phosphorus levels” using red highlight for additions at line 113-119.
Q3. What was the amount of proteins loaded on the acrylamide gels?
Response:After proteins extraction, BCA Kit was used to measure the concentration of various samples. We used 50ug proteins loaded on the acrylamide gels for WB assays. This is added in “2.5 Western blot detection” using red highlight for additions at line 155-157.
Q4. What were the post-hoc tests used for the one- and two-way ANOVA?
Response:We used Bonferroni tests for the one- and two-way ANOVA.
Q5. What does it mean ‘a kind of vitamin D analogs’ (line 403).
Response:Paricalcitol is a selective activator of vitamin D receptor. As the characteristic of paricalcitol is like active vitamin D, we used “a kind of vitamin D analogs” to describe it. To define it exactly, we revised “a kind of vitamin D receptor activator” using red highlight for additions at line 417.
Reference
[1] Yu S, Bruce D, Froicu M, et al.Failure of T cell homing, reduced CD4/CD8alphaalpha intraepithelial lymphocytes, and inflammation in the gut of vitamin D receptor KO mice.[J].Proceedings of the National Academy of Sciences of the United States of America, 2008, 105(52).DOI:10.1073/pnas.0808700106.
[2] Li X, Cai L, Xu H, et al.Somatostatin regulates NHE8 protein expression via ERK1/2 MAPK pathway in DSS induced colitis mice[J].Am J Physiol Gastrointest Liver Physiol, 2016, 311(5):ajpgi.00239.2016.DOI:10.1152/ajpgi.00239.2016.
[3] Xu H, Li J, Chen H, et al. NHE8 Deficiency Promotes Colitis-Associated Cancer in Mice via Expansion of Lgr5-Expressing Cells[J]. Cellular and Molecular Gastroenterology and Hepatology, 2019, 7(1).DOI:10.1016/j.jcmgh.2018.08.005.
[4] Chang, Liu, Hua, et al. NHE8 plays an important role in mucosal protection via its effect on bacterial adhesion.[J].American Journal of Physiology: Cell Physiology, 2013, 305(1):C121-C128.DOI:10.1152/ajpcell.00101.2013.
[5] Guo, Y., et al., Vitamin D receptor involves in the protection of intestinal epithelial barrier function via up-regulating SLC26A3. 655 J Steroid Biochem Mol Biol, 2023. 227: p. 106231.https://doi.org/10.1016/j.jsbmb.2022.106231.

Reviewer 2 Report
Comments and Suggestions for Authors
Regarding the article titled: Compromised NHE8 Expression Is Responsible for Vitamin D-deficiency Induced Intestinal Barrier Dysfunction, I continue with the following considerations:
- update the references in the introduction on the characterization of colitis
- indicate the sex of the animals used
- how old were the Caco2 cells after trypsinization and before treatment with TNF-a?
- In Figure 1, block B, which contains the animals' colons, is very bright, making it difficult to identify the measurement,
- Figure 2, indicates a scale in item D
- Indicate limitations of the study
- fix notation for Latin words and revise, some words are missing letters, like 3D-orgnoid.
Author Response
Dear reviewer,
Re: Manuscript Number: nutrients-2684807 and Title: Compromised NHE8 Expression Is Responsible for Vitamin D-deficiency Induced Intestinal Barrier Dysfunction.
Thank you for your comments concerning our manuscript. Those comments are valuable and very helpful. We have read through comments carefully and have made corrections. Revisions in the text are shown using red highlight for additions. The responses to your comments are presented following. Please see the attachment.
We highly appreciate your time and consideration.
Sincerely,
Corresponding author:
Chunhui Wang
E-mail: wangch@scu.edu.cn
Xiao Li
E-mail: lixiao5166@hotmail.com
Reviewer 2
Regarding the article titled: Compromised NHE8 Expression Is Responsible for Vitamin D-deficiency Induced Intestinal Barrier Dysfunction, I continue with the following considerations:
Q1. Update the references in the introduction on the characterization of colitis.
Response:We updated the newest references on the characterization of colitis.
Q2. Indicate the sex of the animals used
Response:We added more detailed information in “2.1. Experimental animal models” using red highlight for additions at line 92-93.
Q3. How old were the Caco2 cells after trypsinization and before treatment with TNF-a?
Response:Caco2 cells in this study were all treated in 4-10 passages.
Q4. In Figure 1, block B, which contains the animals' colons, is very bright, making it difficult to identify the measurement.
Response:We corrected the picture in Figure 1B.
Q5. Figure 2, indicates a scale in item D
Response:We corrected the picture in Figure 2D
Q6. Indicate limitations of the study
Response:We supplemented “Limitation” in our study using red highlight for additions at line 503-509.
Q7. Fix notation for Latin words and revise, some words are missing letters, like 3D-orgnoid.
Response:We revised the wrong-spelling words in our study.

Reviewer 3 Report
Comments and Suggestions for Authors
Dear Authors,
Congratulations on the article, which was prepared in a transparent and readable way thanks to the enormous amount of laboratory work put into it.
Due to my obligation as a reviewer, I only have minor editorial comments regarding the "Material and methods" section, the completion of which will increase the practical value of the article.
Line 104-105: Due to significant differences between manual and automatic ELISA determinations, please provide information about which protocol was used in the described research. How many repetitions of a given sample were performed?
Line 106: What biochemical assay? The machine and company are missing
Line 110-111: Was only trypsin used, or also trypsin inhibitor after cell detachment? If so, this information is missing.
Line 107-121: There is no information on how many times a given culture has been repeated. The results of the experiment are the result of how many repetitions?
Sincerely yours,
Reviewer
Author Response
Dear reviewer,
Re: Manuscript Number: nutrients-2684807 and Title: Compromised NHE8 Expression Is Responsible for Vitamin D-deficiency Induced Intestinal Barrier Dysfunction.
Thank you for your comments concerning our manuscript. Those comments are valuable and very helpful. We have read through comments carefully and have made corrections. Revisions in the text are shown using red highlight for additions. The responses to your comments are presented following. Please see the attachment
We highly appreciate your time and consideration.
Sincerely,
Corresponding author:
Chunhui Wang
E-mail: wangch@scu.edu.cn
Xiao Li
E-mail: lixiao5166@hotmail.com
Reviewer 3
Q1. Due to significant differences between manual and automatic ELISA determinations, please provide information about which protocol was used in the described research. How many repetitions of a given sample were performed?
Response:We have supplemented detailed information of ELISA in “2.2. Determination of serum 25(OH)2VitD3, calcium and phosphorus levels” using red highlight for additions at line 113-119. Three repetitions were performed for each sample per experiment.
Q2. What biochemical assay? The machine and company are missing
Response:The serum calcium and phosphorus levels were detected by an automatic biochemical an-alyzer (Hitachi, China). We have added detailed information in “2.2. Determination of serum 25(OH)2VitD3, calcium and phosphorus levels” using red highlight for additions at line 118-119.
Q3. Was only trypsin used, or also trypsin inhibitor after cell detachment? If so, this information is missing.
Response:To guarantee the adequate suspension of Caco-2 cells during the passage, we used trypsin to digest cells. After cells falling off from incubator bottom, the enzymatic digestion was neutralized with an equal volume of media with 10% FBS. We did not use trypsin inhibitor after cell detachment.
Q4. There is no information on how many times a given culture has been repeated. The results of the experiment are the result of how many repetitions?
Response:Caco2 cells in this study were all treated in 4-10 passages. Data in this study were obtained from three independent trials performed in triplicate. We have supplemented these information using red highlight for additions at line125-126, 193-194.

Round 2
Reviewer 1 Report
Comments and Suggestions for Authors
The authors replied to the various comments.
please remove « a kind ». Paricalcitol is a VDR activator.
Author Response
Dear reviewer,
Thanks for your advice. We have deleted the phrase "a kind of" at line 417.
Sincerely,
Corresponding author:
Chunhui Wang
Xiao Li